

# Patients with chronic periodontitis are more likely to develop upper urinary tract stone: a nation-wide population-based eight-year follow up study

I-Shen Huang[1,2,3,4], Sung-En Huang[5], Wei-Tang Kao[6,7], Cheng-Yen Chiang[8], To Chang[9], Cheng-I Lin[10], Alex T. Lin[2,3,4], Chih-Chieh Lin[3], Yu-Hua Fan[2,3] and Hsiao-Jen Chung[2,3,4]

[1] Department of Physiology, School of Medicine, National Yang-Ming University, Taipei, Taiwan
[2] Department of Urology, School of Medicine, National Yang-Ming University, Taipei, Taiwan
[3] Department of Urology, Taipei Veterans General Hospital, Taipei, Taiwan
[4] Department of Urology, School of Medicine, Shu-Tien Urological Research Center, National Yang-Ming University, Taipei, Taiwan
[5] Hsinchu City, Taiwan
[6] Department of Urology, Shuang Ho Hospital, Taipei Medical University, New Taipei City, Taiwan
[7] Graduate Institute of Clinical Medicine, Taipei Medical University, New Taipei City, Taiwan
[8] Division of Urology, Surgical Department, Taoyuan General Hospital, Ministry of Health and Welfare, Taoyuan, Taiwan
[9] Division of Urology, Surgical Department, National Yang Ming University Hospital, Yilan, Taiwan
[10] Houston, TX, United states of America

Corresponding author
Hsiao-Jen Chung,
hjchung@vghtpe.gov.tw

## ABSTRACT

**Background**. The purpose of this study was to investigate the relationship between chronic periodontitis (CP) and upper urinary tract stone (UUTS) in Taiwan by using a population-based data set.

**Methods**. A total of 16,292 CP patients and 48,876 randomly-selected controls without chronic periodontitis were selected from the National research database and studied retrospectively. Subjects selected have not been diagnosed with UUTS previously. These subjects were prospectively followed for at least eight years. Cox regression models were used to explore the connection between risk factors and the development of UUTS.

**Results**. The CP patients have a greater chance of developing UUTS compared to controls (1761/16292, 10.8% vs. 4775/48876, 9.8%, $p$-values $< 0.001$). Conditioned logistic regression suggested CP increases the risk of UUTS development (HR 1.14, 95% CI [1.08–1.20], $p < 0.001$). After respective adjustment for age, gender, hypertension and diabetes, results showed that CP still increases the risk of developing UUTS (HR 1.14, 95% CI [1.08–1.20], $p < 0.001$).

**Conclusion**. By using a population-based database with a minimum eight 8 follow-up of CP in Taiwan, we discovered patients with CP are more likely to develop UUTS.

## INTRODUCTION

Urolithiasis is a common urological disease with prevalence ranging from 5% to 20%, that varies with gender, age, race and geographical region (*Scales et al., 2012*; *Hall, 2009*; *Stamatelou et al., 2003*). The incidence has increased in the US in the last three decades and for in Taiwanese population there has been an increase of 8.9% in men between 1998 to 2010 (*Huang et al., 2013*). Recurrence rate was 6–10% within 1 year and up to 35–50% during a 5 year follow up (*Huang et al., 2013*; *Pearle et al., 2005*; *Penniston et al., 2011*). The high prevalence and recurrence rate represent a significant cost burden to the healthcare system, especially for the population in their 40s to 50s, where the annual expenditure is estimated to exceed $5 billion, including direct (treatment) and indirect (work productivity) cost (*Penniston et al., 2011*; *Saigal et al., 2005*).

Chronic periodontitis (CP) is another prevalent inflammatory disease of tooth-supporting tissue, caused by oral bacterial infection and resulting in dysregulation of host immuno-inflammatory response. In the United States, adults aged 30 years or older have an estimated prevalence of 45% and CP is positively correlated with increased age and male gender (*Eke et al., 2012*). In CP, action of matrix degrading enzymes and lipopolysaccharide from oral bacteria trigger the aggregation of an interleukin and tumor necrosis factor, which amplifies the inflammatory response and reduces the damage repair ability of fibroblast, leading to osteoclastogenesis activation and subsequent alveolar bone damage (*Armitage & Science and Therapy Committee of the American Academy of Periodontology Research, 2003*; *Schenkein, 2006*). In addition, during the progression and development of CP, bone metabolism and bone remodeling are altered and the marker of bone metabolism could be used as reflection of its progression (*Yoshihara et al., 2009*). In patients with severe periodontitis, the levels of parathyroid hormones (PTH) were higher and levels of 25(OH)D were lower, indicating that compared to normal subjects, CP patients may have more serious bone resorption and demineralization (*Schulze-Späte et al., 2015*).

Due to elevated PTH in severe CP patients, which possibly brings hypercalcemia and clinical symptoms such as urolithiasis, we hypothesize that CP can potentially induce urolithiasis. This is this first reported study exploring the relationship between CP and urolithiasis. We analyzed the connection between initial chronic periodontitis and subsequent risk of urolithiasis during the 8-year follow up period, with a nationwide population database.

## MATERIALS AND METHODS

The study was performed according to the Taipei Veterans General Hospital Institutional Review Board approved protocol (TPEVGH IRB number: 2018-03-003CC).

### Data source

The National Health Insurance (NHI), launched by the department of Health in Taiwan, has prospectively collected original data for reimbursement, and, since the end of 1996, over 95% of the Taiwanese population are receivers of health care services. The NHI
performs validations of medical charts to ensure the accuracy of documented diagnoses. The official longitudinal dataset derived from the Taiwan NIH, Longitudinal Health Insurance Database 2000 (LHID 2000) contains all of the original claim data of 1,000,000 beneficiaries enrolled in year 2000, randomly sampled from the year 2000 Registry for Beneficiaries in the NHI research database. The LHID 2000 has no record of statistically significant difference in gender distribution, age or healthcare cost between the enrollees from the LHID 2000 and the original NHI research database.

## Selected cases and controls

From 1997 to 2001, a total of 371,185 subjects without previous diagnosis of urinary stones (ICD-9-CM codes 592 "calculus of the kidney and ureter", 592.0 "calculus of the kidney", 592.1 "calculus of the ureter" or 592.9 "urinary calculus, unspecified" and chronic periodontitis (ICD-9-CM codes 523.4 "chronic periodontitis", 523.5 "periodontitis", 523.8 "other specified periodontal diseases" or 523.9 "unspecified gingival and periodontal disease") were enrolled for a minimal follow-up of 8 years. Of the study population, 16,292 subjects aged 30–70 years (8,237 males and 8,055 females) developed chronic periodontitis during the follow-up period and served as the study group. The definition of upper urinary tract stones (UUTS) was present in the ICD-9-CM codes: 592.0, 592.1 or 592.9 at least twice, whereas the definition of chronic periodontitis was identified as ICD-9-CM code 523.4 with the additional procedure codes 91006C, 91007C, 91009B or 91010B.

Proven risk factors of urinary stones, such as hypertension (ICD-9 codes 401–405) and diabetes mellitus (ICD-9 codes 250), were included into the multi-variable analysis. To increase the validity of our study, subjects with a diagnosis of hypertension and/or diabetes mellitus for at least three times within one year were enrolled.

A comparison cohort without chronic periodontitis during the same period was randomly selected, frequency matched 1:3 ($n = 48,876$) for gender, age and comorbidities (diabetes, hypertension, hyperlipidemia). By the end of 2009, all subjects were followed up for a minimum of 8 years.

## Statistical analyses

The distribution of demographic status and comorbidities, including age, sex, diabetes and hypertension, were compared between the CP and non-CP controls. Continuous variables were evaluated by student's $t$ test, and categorical variables were analyzed by the Pearson chi-square test. We compared the study population's characteristics by a crude model to match sex and age, and the crude model was, then, further used again in the adjusted model to match the study population with hypertension and diabetes.

Results are represented as hazard ratio (HR) with a corresponding 95% confidence interval (95% CI). Cox frailty proportional hazard regression models were used to determine the connection between risk factors and the development of upper urinary tract stones. The UUTS-free survival probability in patients regardless of CP matching or hypertension and diabetes during the follow up period was estimated using the Kaplan–Meier method. A log-rank test was applied to generate figure data. We used SAS 9.2 statistics software (SAS Institute Inc., Cary, NC, USA) to analyze data and statistical significance was considered by a conventional $p$ value <0.05.

**Table 1 Demographic characteristics of patients with chronic periodontitis and controls.**

| | Patients with CP (n = 16292)- | | Controls (n = 48876) | | |
|---|---|---|---|---|---|
| Variable | Total No. | % | Total No. | % | P value |
| Sex | | | | | 1.00 |
| Male | 8,237 | 50.6% | 24,711 | 50.6% | |
| Female | 8,055 | 49.4% | 24,165 | 49.4% | |
| Age | | | | | 0.634 |
| <31 | 526 | 3.2% | 1,840 | 3.8% | |
| 31–40 | 6,592 | 40.5% | 19,654 | 40.2% | |
| 41–50 | 5,641 | 34.6% | 16,451 | 33.7% | |
| 51–60 | 2,360 | 14.5% | 7,239 | 14.8% | |
| 61–70 | 1,173 | 7.2% | 3,692 | 7.6% | |
| Personal history | | | | | |
| Diabetes | 2,870 | 17.6% | 7,782 | 15.9% | <0.001 |
| Hypertension | 4,870 | 29.9% | 14,983 | 30.7% | 0.067 |

## RESULTS

The demographic and clinical characteristics of the 65,168 subjects and controls, including 16,292 patients with CP and 48,876 control subjects, are shown in Table 1. Both groups had a similar distribution of sex and age, and were predominantly male (50.6%), 31–40 years (40.5 years) of age. Diabetes was more prevalent in the CP group at baseline (17.6% versus 15.9%, $p < 0.001$). Overall, after a minimal follow-up of at least eight years, analysis demonstrates CP subjects have a greater probability percentage for further development of UUTS than the controls (Hazard ratio 1.14, 95% confidence interval 1.08–1.20, $p$-values <0.001), specifically, 1,761 (10.8%) of the study subjects, and 4,775 (9.8%) of the controls developed UUTS (Table 2). Table 3 represents the association between UUTS and the subjects' characteristics. Age, sex, diabetes and hypertension were both associated with developing UUTS. (all $p$-values <0.001). After adjusting for age, gender, hypertension and diabetes, the Cox regression model fitted, based on the study subjects and controls, showed that patients with CP still possess a significantly increased risk of developing UUTS. (HR = 1.14, 95% CI [1.08–1.20], $p < 0.001$) (Table 4). A Kaplan-Meier analysis showed that, during the 13-year study period, the overall cumulative incidence of UUTS was 19.2% higher for patients with CP than for those who did not have CP ($p < 0.001$, Fig. 1).

## DISCUSSIONS

In this nationwide population-based retrospective cohort study, a greater percentage of subjects with CP developed UUTS than that without chronic periodontitis. After adjusting for hypertension and diabetes, regression analysis still pointed to CP as an independent factor for UUTS.

To date there have been no studies that explore the association between CP and UUTS. It appears that these two diseases share metabolic syndrome (MetS) as a common risk factor, which includes a spectrum of the following medical conditions: dysglycemia, dyslipidemia,

**Table 2 Crude HR for new-onset upper urinary tract stone (UUTS) among patients with chronic periodontitis and comparison controls.**

| New-onset UUTS | Total sample (n = 65,168) | | Patients with CP (n = 16,292) | | Comparison controls (n = 48,876) | |
|---|---|---|---|---|---|---|
| 8-year follow up | No. | % | No. | % | No. | % |
| Yes | 6,536 | 10.0 | 1761 | 10.8 | 4,775 | 9.8 |
| No | 58,632 | 90.0 | 14,531 | 89.2 | 44,101 | 90.2 |
| Crude HR (95% CI) | – | | 1.14 | | 1.00 | |

**Table 3 Demographic characteristics of subjects with and without new-onset upper urinary tract stone (UUTS) during follow-up.**

| Variable | Subjects with UUTS (n = 6,536) | | Subjects without UUTS (n = 58,632) | | P value |
|---|---|---|---|---|---|
| | Total No. | % | Total No. | % | |
| Age | Mean 44.8 ± 9.9 | | Mean 43.5 ± 9.7 | | <0.001 |
| Sex | | | | | <0.001 |
| Male | 4,107 | 62.8 | 28,841 | 49.2 | |
| Female | 2,429 | 37.2 | 29,791 | 50.8 | |
| Periodontitis | | | | | <0.001 |
| Yes | 1,761 | 26.9 | 14,531 | 24.8 | |
| No | 4,775 | 73.1 | 44,101 | 75.2 | |
| Diabetes | | | | | <0.001 |
| Yes | 1,379 | 21.1 | 9,273 | 15.8 | |
| No | 5,157 | 78.9 | 49,359 | 84.2 | |
| Hypertension | | | | | <0.001 |
| Yes | 2,591 | 39.6 | 17,262 | 29.4 | |
| No | 3,945 | 60.4 | 41,370 | 70.6 | |

**Table 4 The risk for upper urinary tract stone among chronic periodontitis, hypertension and diabetes patients in Cox proportional hazard regression.**

| | Crude HR | 95% CI | P value | Adjusted HR | 95% CI | P value |
|---|---|---|---|---|---|---|
| Main effect | | | | | | |
| Chronic periodontitis | 1.14 | 1.08–1.20 | <0.001 | 1.14 | 1.08–1.20 | <0.001 |
| Comorbidities | | | | | | |
| Hypertension | 1.35 | 1.28–1.44 | <0.001 | 1.17 | 1.10–1.25 | <0.001 |
| Diabetes | 1.48 | 1.41–1.55 | <0.001 | 1.42 | 1.34–1.49 | <0.001 |

Notes.
Crude HR, relative hazard ratio; Ajusted HR, multivariable analysis including age, sex and cormorbidities.

visceral obesity, and hypertension. In longitudinal and cross-sectional studies, there is a positive correlation between MetS and CP—as the number of MetS components in an individual increases, so does the risk of periodontitis (*Kaye et al., 2016*; *Lee et al., 2015*). The postulated mechanism for this association is well studied with dysglycemia, a key feature of MetS. Enhanced production of irreversible advanced glycation end product (AGE)

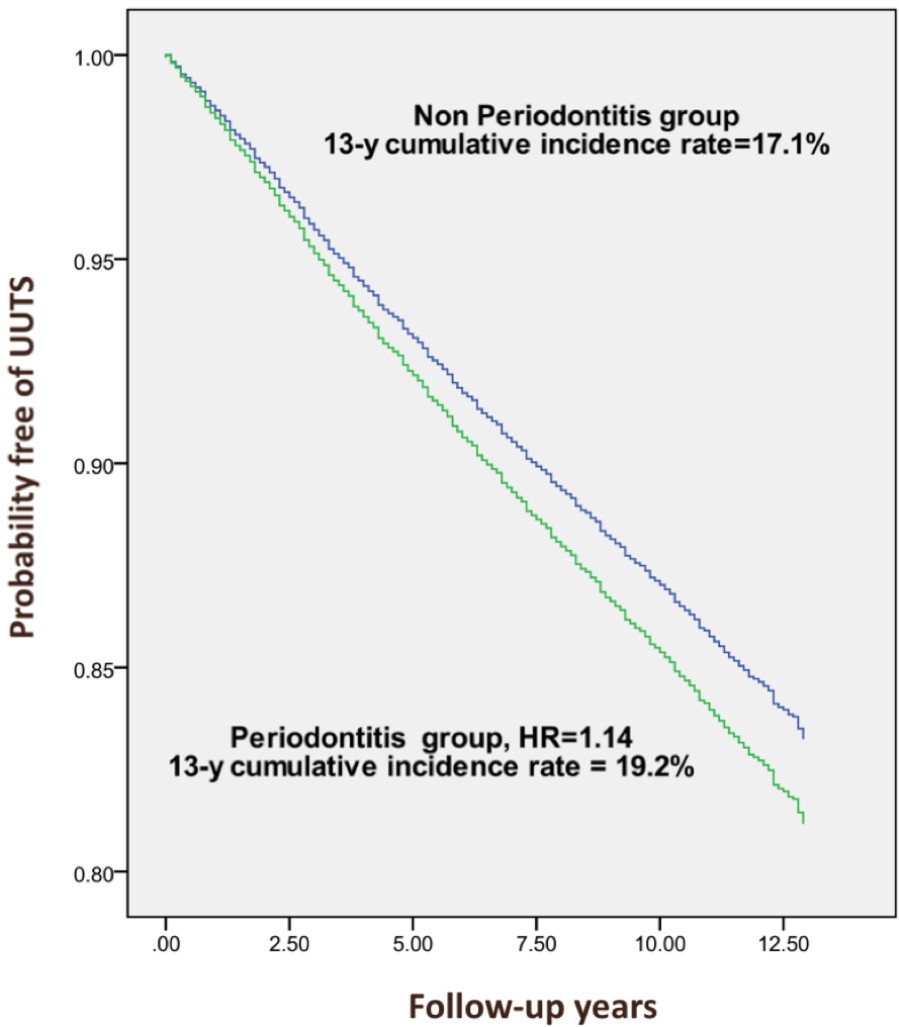

**Figure 1** Probability free of upper urinary tract stone (UUTS) for periodontitis and non-periodontitis patients.

in a hyperglycemia state may result in a defective constitution of extracellular matrix components, resulting in compromised tooth supporting tissue, devastated apparatus and eventual teeth exfoliation (*Gurav, 2013*). In addition, through the induction of IL-1b, TNF-a and PGE2 in dysglycemic state, increased oxidative stress may be associated with an increased expression of the receptor activator of nuclear factor kappaB ligand (RANKL), which plays a critical role in periodontal bone resorption (*Wu, Xiao & Graves, 2015*). In a meta-analysis study studying MetS and urolithiasis, results showed that a higher prevalence of urolithiasis tended to exist among patients with three or more MetS traits (*Wong et al., 2016*), which is similar to the relationship between MetS and periodontitis. Interestingly, a higher prevalence of uric acid stones and a lower prevalence of calcium phosphate stones tended to exist among patients who have two or more MetS factors, but the relative frequency of calcium oxalate stones remained regardless of the number of

MetS factors (*Kadlec et al., 2012*). A higher frequency of uric acid stones in MetS patients can possibly be explained as consequence of acidic urine, caused by impaired ammonia excretion and increase of endogenous acid production, secondary to insulin resistance (*Strohmaier, Wrobel & Schubert, 2012*).

However, while CP and urolithiasis are both associated with MetS, especially hyperglycemia and hypertension (*Wong et al., 2016*), after adjusting for diabetes and hypertension, CP remains an independent risk factor for developing UUTS. Due to the wide spectrum of endocrine and urine abnormalities in urolithiasis patients, it may be more rational to investigate the abnormalities found in chronic periodontitis patients and correlate with pathophysiology of urolithiasis. Given that periodontitis is characterized by tissue and bone destruction, serum calcium level, bone metabolism and, specifically, bone remodeling are altered during the disease progression (*Yoshihara et al., 2009*; *Schulze-Späte et al., 2015*). In a recent study, *Amarasena et al. (2008)* evaluated 266 Japanese subjects aged 70 years and assessed the relationship of baseline serum calcium and the periodontal disease progression by a follow up of six consecutive years. The finding indicated that subjects with a low serum calcium level have a significantly higher chance of developing periodontal disease. This is corroborated in the Buffalo OsteoPerio study, which followed the periodontal condition in 1,025 postmenopausal women for 5 years, and concluded that women with severe periodontitis and osteoporosis may exhibit accelerated bone loss over time (*LaMonte et al., 2013*). A higher rate of recurrent periodontitis can also be expected in subjects with osteoporosis after periondontal treatment compared to individuals with normal skeletal bone marrow density (BMD) (*Gomes-Filho et al., 2013*). Nonetheless, in another observational, prospective study evaluating men 65 years and older, although lower vitamin D and higher level of PTH were observed in severe periodontitis patients, these measures were not associated with progression. Additionally, bone metabolism markers were not associated with periodontis severity, but men with baseline lower levels of bone remodeling markers tended to have a greater improvement in periodontitis (*Schulze-Späte et al., 2015*). On the other hand, the pathophysiological mechanisms for urolithiasis are complex and diverse, including hypercalciuria, hyperuricosuria, hypocitraturia, hyperoxalateuria and abnormalities in urine pH (*Pak, 1991*). The mineral composition of stones is calcium oxalate or calcium phosphate in 80% of urolithiasis patients, whereas uric acid, struvite and cystine stone represent the rest. Formation of the crystal stone can happen from supersaturation of these crystalline material in urine, which is common when daily urine output is less than 2 liters (*Ratkalkar & Kleinman, 2011*). Hypercalciuria is the most common abnormal finding among calcium urolithiasis patients, found in 30–60% of them, and may result in supersaturation of urinary calcium salts (*Pak et al., 1980*). In hypercalciuric kidney stone for mers, osteoporosis, by means of $T$-score $<-2.5$, is a prevalent finding. Moreover, hypercalciuria is prevalent in postmenopausal women, and also represents an important predictor for low bone mass (*Giannini et al., 2003*). Altogether, bone loss or presence of low bone mineral density is a link both connected to PD and urolithiasis, which could be the explanation found for PD increasing the chance of further UUTS, but this hypothethized biological model needs to be examined by further

prospective study evaluating bone mineral density in PD patients with an occurrence of urolithiasis. The mechanism for this finding is, however, beyond the scope of this paper.

In our study, we only enrolled patients between 30–70 years old, which is the vast majority of the working population in Taiwan. Therefore, these findings cannot be extrapolated to subjects in different age ranges. Male to female ratio is 1.69. In further UUTS occurrence in our study, this ratio is higher in correspondence with previous Taiwanese population studies, in which the ratio is 1.55 after age adjustment, and also greater than that for the US population reported (*Huang et al., 2013*; *Penniston et al., 2011*).

Although this study detected an association of chronic periodontits with subsequent UUTS development, aided by the use of national database, our results need to be carefully intepreted by knowing their limitations. First, the data were obtained from the NHI database; therefore, certain patient information, including blood pressure and serum glucose, cannot be obtained, which makes the diagnosis less accurate than those made through standardized procedures. Second, other confounding factors such as personal health behavior, diet, lifestyle, height, weight, waistline measurements, drinking and cigarette smoking could not be adjusted in the NHIRD administrative data set. Lastly, we identified patients by diagnostic code (ICD-9), and increased the accuracy of the UUTS diagnosis by the presence of the ICD code twice. The accuracy of chronic periodontitis diagnosis was increased by additional procedure codes. However, we cannot evaluate the severity of disease and the composition of UUTS. Further prospective randomized study may be needed to elucidate the findings of this study and investigate the cause of UUTS in patients with chronic periodontitis.

## CONCLUSION

By using a population-based database, with a minimum 8 year follow-up, of chronic periodontitis in Taiwan, we discovered patients with CP are likely to develop UUTS. CP patients receiving treatment for periodontitis should also be told about its connection with further development of UUTS.

### Funding
The authors received no funding for this work.

### Competing Interests
The authors declare there are no competing interests.

### Author Contributions
- I-Shen Huang conceived and designed the experiments, performed the experiments, analyzed the data, contributed reagents/materials/analysis tools, prepared figures and/or tables, authored or reviewed drafts of the paper.
- Sung-En Huang, Wei-Tang Kao, Cheng-Yen Chiang, To Chang, Chih-Chieh Lin and Yu-Hua Fan analyzed the data.

- Cheng-I Lin conceived and designed the experiments.
- Alex T. Lin conceived and designed the experiments, performed the experiments, analyzed the data.
- Hsiao-Jen Chung conceived and designed the experiments, performed the experiments, analyzed the data, contributed reagents/materials/analysis tools, prepared figures and/or tables, authored or reviewed drafts of the paper, approved the final draft.

## Human Ethics

The following information was supplied relating to ethical approvals (i.e., approving body and any reference numbers):

The study was performed according to the Taipei Veterans General Hospital approved Institutional Review Board protocol (IRB number: 2018-03-003CC).

## Data Availability

The raw data are provided in a Supplemental File.

## Supplemental Information

Supplemental information for this article can be found online at http://dx.doi.org/10.7717/peerj.5287#supplemental-information.

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
