# Peer review of "Patients with chronic periodontitis are more likely to develop upper urinary tract stone: a nation-wide population-based eight-year follow up study"

_PeerJ, doi:10.7717/peerj.5287_

## Round 0.1 · original submission · Minor Revisions

All reviewers commented on the English language. You must attend to the grammatical errors and other issues raised by the reviewers and substantially edit the document before any resubmission. As noted by Reviewer 1 we suggest you seek the services of an editing company

In addition, there are comments by all 4 reviewers which should be attended to. In particular, the comments of Reviewer 4 and Reviewer 3 are crucial. The statistics must be carefully addressed, and the conclusions need to be appropriately stated.

·

Basic reporting

The article is well written,unambiguous and falls well within the scope of the journal. However, there are several typo and grammatical errors in the manuscript. Language edited certificate from Class'A' language editing company is recommended before publishing the article.The study is presented and published as an abstract abstract i in the Journal of Urology (2013)( http://www.jurology.com/article/S0022-5347(13)02278-7/abstract) as a conference abstract and can be mentioned in the manuscript.

Experimental design

the experimental design is fine.

Validity of the findings

The observations are supported

Additional comments

Please see some of the grammatical and typographical errors in the tracked manuscript. It is better to get it edited for clarity.

Reviewer 2 ·

Basic reporting

Needs language editing
Table 2 seems confusing...need more clarification

Experimental design

well defined research with sufficient detail

Validity of the findings

Discussion part on relationship between chronic periodontitis & UUTI needs to be clearly stated.

Additional comments

needs language editing

·

Basic reporting

Investigators evaluated the association between chronic periodontitis (CP) and kidney stones (UUTS) and report that CP increases the risk of kidney stones significantly independent of key covariates such as age, gender, and diabetes etc.

Experimental design

This is an 8-year follow-up study embedded within a national insurance database in Taiwan. The exposed group is the ones with ICD codes for CP (N=16,292). Controls were randomly selected (do not know how exactly) subjects without CP (N=48,876). Over a roughly 8 year follow-up, the risk of UUTS was 1% higher in the CP group which was statistically significant (HR=1.14, 95% CI, 1.08–1.20, p < 0.001) which remained significant after adjusting for age, gender, hypertension, and diabetes.

Validity of the findings

The study is well conducted and is large. The analysis is appropriate. The results are presented clearly and logically.

Are the methods used to detect the absence of CP in the 'controls' is the same as that used for the CP group obtained from the insurance data? That can influence the findings.

There is only a 1% increase in UUTS and that can very well be due to chance as the study involved thousands. This needs to be discussed. A strong biological model how the CP might increase UUTS is also a worthy discussion.

However, the external validity of the study is supported by the findings that the CP group had significantly higher diabetes as expected.

Additional comments

The first word in the abstract is a typo. Careful review and revision would enhance the quality of the manuscript.

·

Basic reporting

This manuscript needs to be edited by an individual fluent in the English language.
Literature references don't give full support background to hamper the research question.

Experimental design

The research question is skewed. While the authors justify, in a sustained way, the association of PTH levels elevation in periodontitis patients, the association of uric acid metabolism with periodontitis is merely an attempt without scientific support. I would suggest the authors put more effort in first to guarantee this association of uric acid metabolite with periodontal disease prior to any further assumption.

The authors didn't consider obesity, hyperuricemia, family history of stone disease and dietary habits as possible confounders, and those are important characteristics to minimize selection and analysis biases.

This manuscript should be reviewed by a biostatistician to ensure the statistical analyses are appropriate. Why did the authors decide for a frequency matched 1:3? Did you perform any statistical test to get score matching to select controls?

Validity of the findings

Although this can be considered a novelty and the robust data, this paper has serious flaws and the authors must consider tune down the statements such as ‘CP increase the risk of developing UUTS’.

Additional comments

I would suggest the authors put more effort in first to guarantee this association of uric acid metabolite with periodontal disease prior to any further assumption. Please respond to the questions referred.

---

## Round 0.2 · accepted · Accept

Dear Dr I-Shen,

Congratulations! Your article has been accepted for publication but I ask you to address the remaining language issues while in production.

Reviewer 2 ·

Basic reporting

Article is more clearly stated and language editing done as stated before

Experimental design

No comment

Validity of the findings

Conclusion is well stated

·

Basic reporting

Authors have addressed the previous concerns and have obtained the help of a Native English speaker to edit the manuscript.

Experimental design

Appropriate.

Validity of the findings

Statistically significant 1% increase in the outcome in CP group may still be due to the large sample size. Is it clinically meaningful?

Additional comments

Adequate revisions.

·

Basic reporting

I acknowledge that the authors improved the english, but this paper still needs professional English assessment.

Experimental design

All experimental questions were answered.

Validity of the findings

Nonthing to declare.

Additional comments

I acknowledge that the authors improved the English, but this paper still needs professional English assessment.